# Hepatitis C Virus Downregulates Core Subunits of Oxidative Phosphorylation, Reminiscent of the Warburg Effect in Cancer Cells

**DOI:** 10.3390/cells8111410

**Published:** 2019-11-08

**Authors:** Gesche K. Gerresheim, Elke Roeb, Audrey M. Michel, Michael Niepmann

**Affiliations:** 1Institute of Biochemistry, Medical Faculty, Justus Liebig University, Friedrichstrasse 24, 35392 Giessen, Germany; 2Department of Gastroenterology, Justus Liebig University, Klinikstrasse 33, 35392 Giessen; Germany; Elke.Roeb@innere.med.uni-giessen.de; 3School of Biochemistry and Cell Biology, University College Cork, Cork, Ireland; audreymannion@gmail.com

**Keywords:** HCV, HCC, hepatocellular carcinoma, fibrosis, oxidative phosphorylation, mitochondrial respiratory chain, NADH-ubiquinone oxidoreductase, cytochrome c oxidase, ATP-Synthase, warburg effect

## Abstract

Hepatitis C Virus (HCV) mainly infects liver hepatocytes and replicates its single-stranded plus strand RNA genome exclusively in the cytoplasm. Viral proteins and RNA interfere with the host cell immune response, allowing the virus to continue replication. Therefore, in about 70% of cases, the viral infection cannot be cleared by the immune system, but a chronic infection is established, often resulting in liver fibrosis, cirrhosis and hepatocellular carcinoma (HCC). Induction of cancer in the host cells can be regarded to provide further advantages for ongoing virus replication. One adaptation in cancer cells is the enhancement of cellular carbohydrate flux in glycolysis with a reduction of the activity of the citric acid cycle and aerobic oxidative phosphorylation. To this end, HCV downregulates the expression of mitochondrial oxidative phosphorylation complex core subunits quite early after infection. This so-called aerobic glycolysis is known as the “Warburg Effect” and serves to provide more anabolic metabolites upstream of the citric acid cycle, such as amino acids, pentoses and NADPH for cancer cell growth. In addition, HCV deregulates signaling pathways like those of TNF-β and MAPK by direct and indirect mechanisms, which can lead to fibrosis and HCC.

## 1. Hepatitis C Virus Replication in the Liver

Infection with Hepatitis C Virus (HCV) is one of the major causes for liver damage. Although HCV can cause acute infection with severe and sometimes fatal outcomes, the main problem with HCV infection is that in about 70% of all infections, the virus establishes chronic replication in the liver [1]. In these cases, HCV manages to escape the innate and adaptive immune responses to allow the virus to replicate in the hepatocytes “under the radar” [2,3]. In such cases, the infection usually remains inapparent and undiagnosed, and such chronic infection over the years often leads to liver fibrosis, cirrhosis and in many cases, finally, to liver cancer (hepatocellular carcinoma, HCC) [4]. Since the liver is quite a soft organ that is functionally largely homogenous and is equipped with a redundant capacity to regulate the body’s metabolite flux requirements in normal conditions, impaired liver function becomes apparent only when the liver is heavily infiltrated by the cancer. Therefore, patients show up with symptoms often very late, resulting in high recurrence rates after surgery and deaths from liver cancer, even after treatment of the HCV infection [4,5,6,7].

HCV comes as a so-called lipo-viro particle (Figure 1A) [8,9]. The particle is a fusion between viral and cellular components, with the involvement of cellular components largely exceeding what is usually recruited by other enveloped viruses. This is due to the close association of HCV’s assembly pathway with the assembly of Very Low-Density Lipoproteins (VLDL) [8,9]. Cellular proteins included in the lipo-viro particles are apolipoproteins (Apo) A-I, B, C-II and E (Figure 1A). The viral genome is covered by the HCV core protein, and the viral envelope contains proteins E1 and E2. Binding of HCV to hepatocytes and its uptake into the cells is conferred by several cellular receptors (comprehensively reviewed in [9,10,11]).

The HCV genome is a single-stranded RNA of about 9.6 kilobases with positive orientation (Figure 1B) [14], i.e. after uncoating, the viral RNA can be directly translated in the cytoplasm of the cell [15]. The single open reading frame (ORF) encodes a polyprotein that is cleaved into the mature gene products by viral and cellular proteases [12,16]. These proteins include the above-mentioned structural proteins as well as non-structural (NS) proteins, including the viral RNA-dependent RNA polymerase (replicase) NS5B. In contrast to cellular mRNAs, the HCV genomic RNA is not capped at its 5′ end, but translation initiation is mediated by an Internal Ribosome Entry Site (IRES) (Figure 1B) [17,18,19,20]. This strategy comes with essentially two advantages. On the one hand, the viral RNA can escape translational downregulation during the cellular innate immune response that affects cap-dependent translation initiation [21]. On the other hand, this strategy avoids the need to have translation regulation signals like a 5′-cap nucleotide and a poly(A) tail at the very 5′- and 3′ends of the viral RNA genome, leaving these ends free for replication signals that allow the viral replicase to initiate genome replication [12,13,22].

Viral proteins produced in the pilot round of translation associate with cellular membranes of the Endoplasmic Reticulum (ER) and induce the formation of an assembly of membrane vesicles, the so-called membranous web [23,24,25] (Figure 2). In these vesicles, viral genome replication and the following assembly of virus particles are spatially coordinated, with the RNA replication complex formed by NS2-NS5B proteins. At the very 3′ end of the viral plus strand genome, the viral replicase NS5B initiates the synthesis of antigenome minus strands. This process is regulated by several RNA sequences and secondary structures that are located at the very 3′ end but also within the polyprotein coding region, mostly in the NS5B coding region [12,13,22,26,27,28]. The minus strand antigenome then serves as the template for the production of excess plus strands that finally are encapsidated into new progeny virus particles in the assembly pathway [8,29]. In addition, the initiation of plus strand RNA synthesis is regulated by RNA signals that reside at the 3′ end of the minus strand [30,31].

HCV RNA genome stability, translation and replication are positively regulated by the liver-specific microRNA (miR)-122 [32,33,34]. miR-122 expression is quite unusual among microRNAs (miRNAs) since it constitutes about 70% of all liver microRNA [35] and is nearly absent from other tissues [36]. Therefore, the liver specificity of HCV replication is considered not only to be due to the combination of a variety of cellular surface receptors that are involved in virus particle attachment and uptake [9,10,11], but also largely due to the intracellular enhancement of HCV replication by miR-122 [32,33,34]. There are five or six binding sites for miR-122 in the HCV genome (depending on genotype) [37]. Two sites are located at the very 5′ end in the 5′UTR [38]. The binding of miR-122 to these sites in the 5′UTR occurs cooperatively [39,40] and has been shown to be involved in three different effector functions, namely overall genome replication [32], translation stimulation [33,41,42,43], as well as RNA genome stabilization against nucleases [34,44]. One additional conserved miR-122 binding site is located in the 3′UTR and two or three other sites are located in the NS5B coding region. Although some studies have investigated binding of miR-122 to these sites and the possible consequences for the control of viral translation and replication [45,46,47], the actual functions of these conserved miR-122 binding sites in the HCV replication cycle are still largely unknown.

**Figure 2 cells-08-01410-f002:**
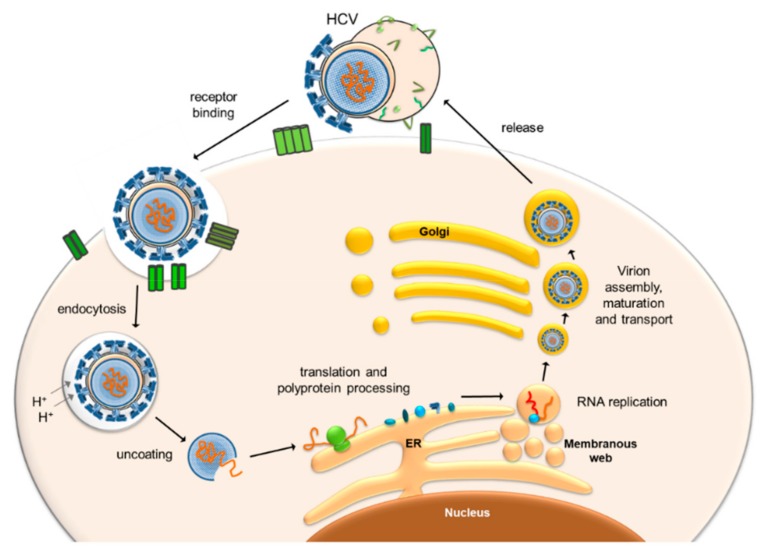
Brief overview over the HCV replication cycle. In the space of Disse in the liver sinusoids, the HCV lipo-viro particle (top) attaches to the basolateral surface of the hepatocyte by virtue of a variety of receptors (not shown in detail here) which include heparan sulfate proteoglycans (HSPGs), the LDL receptor, the scavenger receptor SR-BI, CD81 as well as claudin 1 (CLDN1) and occludin (OCLN) [11]. After endocytosis, pH-dependent fusion with the lysosome triggers uncoating and release of the viral RNA. Following the initial (“pilot”) round of the IRES-mediated translation of the HCV genomic RNA, viral replication proteins recruit membranes from the Endoplasmic Reticulum (ER) to form the closely ER-associated “Membranous web” [23,24,25] which is the site of viral replication. There, new virus particles are formed in close association with the metabolism of lipid droplets (LDs) and VLDLs [8,24,29,48,49,50], and the new viruses are released from the cell.

Compared with some other RNA viruses (e.g. picornaviruses like poliovirus or foot- and-mouth disease virus) [51], HCV does not completely take over the cells’ capacities, resulting in rapid dead-end cell lysis but replicates quite slowly and at a low level [46,52,53]. In this context, it is interesting to note that the HCV IRES element is quite weak compared to cap-dependent translation [54]. In combination with viral counter-measures against the cellular innate immune system and the body’s adaptive immune response [2,55,56], this allows the establishment of long-term ongoing chronic HCV replication, which eventually is a very successful strategy of a virus for “under-cover” spread among host individuals. However, although in most cases, HCV replication in the liver does not lead to complete organ failure within a short time, the virus subtly takes over control of cellular gene expression to promote viral replication [2]. Moreover, the induction of cancer growth in cells that chronically replicate a virus can be regarded as an overall advantage for that virus, since the cells replicating the virus even multiply by themselves and escape the body’s immune system.

Changes in gene expression in the HCV-infected cells that induce fibrosis and HCC not only affect signaling cascades to regulate cell growth, induce the cell cycle and suppress apoptosis [2,57,58], but cancer cell growth also requires reprogramming the metabolism of the cells [59,60]. These changes in metabolism in cancer cells include an enhanced uptake and metabolization of glucose by the main carbohydrate breakdown pathway, glycolysis. However, the consumption of oxygen is slowed down but not completely abolished, while the cells secrete large amounts of lactate. This condition is called “aerobic glycolysis”; it has been described by Otto Warburg [61] and is now known as the “Warburg Effect”. While Otto Warburg considered this effect to be a cause of cancer [61,62], we now know that cancer is induced in cells by changes in gene expression affecting the regulation of growth, apoptosis and cell cycle [59]. However, aerobic glycolysis is a condition that is generally observed in many cancer cells [59]. Consistently with the reprogramming of cancer cells to also grow under low oxygen conditions, HCV replication was shown to be even enhanced at low oxygen pressure [63].

In the following section, we consider the nature of the Warburg Effect and the contribution of HCV replication, resulting in a revised view of the metabolic changes in the Warburg Effect, and we discuss some selected gene expression changes HCV replication applies to the cells in terms of growth and differentiation control.

## 2. The Warburg Effect—Enhanced Glycolysis Flux to Provide Anabolic Metabolites in Cancer Cells

Cancer cells need large amounts of anabolic metabolites for fast growth and cell division. Cancer cells may also need to grow and replicate, even under low oxygen conditions, since regular blood vessels may not be sufficient to provide enough oxygen for their fast growth, while blood vessels newly induced by the tumor still need to grow. For these reasons, cancer cells usually undergo a metabolic switch, which enhances metabolite flux through glycolysis (see Figure 3) towards its end-product pyruvate (Pyr). At the same time, further downstream metabolite consumption by mitochondrial pyruvate dehydrogenase (PDH), citric acid cycle (or tricarboxylic acid [TCA] cycle) and oxidative phosphorylation is reduced but not completely abolished to still allow for sufficient ATP production [64,65,66]. The change to this so-called aerobic glycolysis in tumor cells [61,62] was named the Warburg Effect.

In this condition, enhanced glucose consumption with a bottleneck downstream of pyruvate allows for the withdrawal of considerable amounts of glycolysis intermediates in order to provide metabolites for anabolic purposes. Such metabolites include amino acids for protein synthesis, riboses for nucleotide synthesis, and the reduced form of the anabolic electron carrier nicotinamide-adenine-dinucleotide phosphate (NADPH) which is required for many biosynthesis reactions. Riboses and NADPH are mainly produced from glucose-6-phosphate (G-6-P) in the pentose phosphate pathway (PPP, see Figure 3), while many amino acids are derived from the C_3_ glycolysis intermediates glyceraldehyde-3-phosphate (G-3-P), phosphoenolpyruvate (PEP) and pyruvate. Moreover, the flux of citrate from the mitochondrion to the cytosol is elevated to allow for enhanced biosynthesis of lipids for growth [67]. In contrast to widely accepted assumptions, the activity of PKM2, the pyruvate kinase (PK) isoform that is often expressed in cancer cells, is not rate-limiting but has the highest specific activity of all glycolytic enzymes [68].

The conditions of enhanced metabolite flux through glycolysis in the cancer cells also lead to an excess of the catabolic electron carrier nicotinamide-adenine-dinucleotide (NADH) in the cytosol (Figure 3). Under conditions of limiting oxidative phosphorylation, not all of the cytosolic NADH that is produced in the glyceraldehyde-3-phosphate dehydrogenase (GAPDH) reaction can be reoxidized to NAD^+^ in the mitochondrion. However, the NAD^+^ is required in the cytosol to run the GAPDH reaction; otherwise, metabolite flux through glycolysis would stop. Therefore, the electrons from a fraction of the cytosolic NADH must be disposed of in some other way to allow reoxidation of NADH to NAD^+^. For this reason, the electrons from a fraction of the cytosolic NADH are transferred to pyruvate to yield lactate in the lactate dehydrogenase (LDH) reaction. The lactate is then released from the tumor cells [59,60,61,69] and reused in the liver to produce glucose (known as the “Cori cycle” which is obligatory in the case of muscle working under anaerobic conditions as well as for erythrocytes which do not have mitochondria) [70].

## 3. HCV Reprograms the Metabolism of Infected Cells towards a Cancer-Like State

Although progression of liver cells to histologically detectable fibrosis, cirrhosis and finally, hepatocellular carcinoma, usually takes years [4,58,71,72,73], HCV appears to induce metabolic changes that direct the cell towards the Warburg Effect quite quickly, within a few days or weeks after infection of a cell. In a recent study, we found that the expression of some key components of the mitochondrial respiratory chain complexes were downregulated only 6 days after the start of HCV replication [74]. These key components are the subunits MT-ND1, MT-ND3, MT-ND4, and MT-ND4L of complex I (NADH-ubiquinone oxidoreductase) and MT-CO2 of complex IV (cytochrome c oxidase). MT-ND1, 3, 4 and 4L are core subunits of complex I, which are located directly within the inner mitochondrial membrane and are involved in the enzymatic activity of the complex [75,76]. Similarly, MT-CO2 is a catalytically essential core subunit of complex IV and this subunit is located directly within the inner mitochondrial membrane [77]. These subunits are still encoded in the animal mitochondrial genomes which have essentially—except for genes for mitochondrial ribosomes—lost all protein-coding genes that do not encode largely hydrophobic respiratory chain complex subunits [75,78]. Keeping the genes for these respiratory chain key subunits in the mitochondrial genome was considered to be essential to allow for a short-circuit redox regulation [79]. In an experiment with HCV infected CD8^+^ T cells, in addition to MTND and COX also the F_o_/F_1_ ATP synthase was found to be downregulated at later time points (4 weeks and more) [80,81]. Taken together, these findings show that HCV systematically limits the activity of oxidative phosphorylation (see Figure 3).

Another limitation is the PDH reaction, which is the entrance gate to complete oxidation of the carbohydrate chains in the mitochondrion [66,82,83]. The PDH kinase (PDK) is induced by the transcription factor Hypoxia-inducible factor 1 alpha (HIF-1α), and HIF-1α in turn is induced during HCV replication [84,85]. HIF-1α is normally hydroxylated under normoxic conditions and then degraded, whereas under hypoxic conditions, HIF-1α regulates the response of the cell to cope with the hypoxic conditions [86]. The degradation of HIF-1α under normoxic conditions involves the ubiquitin ligase VHL subunit, which is related to von Hippel-Lindau disease [87].

Upstream of the PDH bottleneck reaction, HCV activates enzymes of glycolysis and the pentose phosphate pathway (Figure 3). Consequently, higher metabolite concentrations were found in HCV-infected subjects, including lactate, pyruvate and amino acids [88]. Both HIF-1α and the proto-oncogene c-MYC were expressed at significantly higher levels in HCV-infected human liver and hepatocytes than in uninfected controls, and HIF-1α and c-MYC, in turn, induce the expression of several glycolysis key enzymes [84,85]. These glycolytic key enzymes are glucokinase (GK), phosphofructoinase-1 (PFK-1) and pyruvate kinase (PK) which together, control metabolite flux through the glycolysis. In addition, the expression of hexokinase 2 (HK2) is upregulated [64] and the activity of hexokinase is increased by its interaction with HCV protein NS5A [89]. In addition, HCV activates the phosphatidyl-inositol-3-kinase (PI3K) - Akt - mammalian target of rapamycin (mTOR) pathway [90,91,92,93] that is usually activated by growth hormones in the presence of sufficient amino acid levels. In particular, HCV translation is upregulated by this pathway [94,95].

In addition, microRNAs (miRNAs) contribute to the upregulation of glycolytic enzymes. HIF-1α mRNA is a direct target of microRNA-199a (miR-199a). miR-199a itself binds to the HCV RNA genome [96,97]. Since miR-199a is not very abundant in cells [36], the several thousand HCV genomes replicating in the infected cell [12,46,53] can be considered to sequester miR-199a and thus, withdraw it from the HIF-1α 3′UTR, leading to upregulation of HIF-1α. Moreover, the PKM2 mRNA is a direct target of miR-122 [98]. This microRNA binds to five or six sites (depending on the genotype) in the HCV genome [32,37,47]. Therefore, HCV is also considered to sequester miR-122 to a considerable extent [46], even though several tens of thousands molecules miR-122 are present in the hepatocyte [35].

In addition to glycolytic enzymes, the enzymes of the pentose phosphate pathway are also upregulated during HCV infection and in hepatocellular carcinoma related to HCV [99,100,101]. The elevated expression of pentose phosphate pathway enzymes in HCC is a good indicator for enhanced metabolite flux towards riboses and NADPH for nucleotide and lipid biosynthesis pathways, as well as enzymes involved in the biosynthesis of glutathione [100,101]. The TCA cycle and oxidative phosphorylation were also found to be largely upregulated a short time after infection, whereas after a longer infection, these pathways essentially changed back to normal or lower activity, while glycolysis and PPP enzymes remained upregulated [99,100]. Consistently, two recent studies found that oxidative phosphorylation is rather downregulated several days after HCV infection [74,80].

HCV promotes this metabolic reprogramming through expression of SQSTM1 and thus, induces the PPP enzymes. SQSTM1/p62 is a protein that is involved in regulation of autophagy, the oxidative defense system [102] and nutrient sensing and inflammation [103]. Since SQSTM1 is upregulated in hepatoma cells 6 days after the beginning of HCV replication [74], we conclude that SQSTM1 may be one of the key regulators that induce metabolic reprogramming in HCV infected cells towards the development of hepatocellular carcinoma. Synergistically, alcohol abuse predisposes individuals to the development of HCC and heightens HCC risk in patients infected with HCV [104,105].

Moreover, glutaminolysis is activated in HCV-infected HCC cells [99,100,101], likely since the NH2 groups released from glutamine are used for incorporation into amino acids during cell growth, while the carbon backbone of glutamine can be used as a carbon source for cell growth when glutamate is used to replenish the TCA cycle by the anaplerotic aminotransferase reaction producing 2-oxoglutarate [99,100]. The glutamate derived from the glutamine is also used for the production of glutathione which is required to combat oxidative stress. Consistently, Glutamate-Cystein ligase is also activated in order to provide more glutathione [101].

In contrast to the above findings that suggest a stimulation of glycolysis in HCV-infected hepatocellular carcinoma cells, other studies, including many studies using animal models and analyzing the state of human HCV patients, showed that HCV infection causes insuline resistance in the HCV-infected cells, leading to elevated glucose levels in the blood and a prediabetic or diabetic state of the patient (reviewed in [106,107,108]). In this condition, intracellular key enzymes of glucose synthesis were found to be upregulated, like phosphoenolpyruvate carboxykinase (PEPCK) and glucose-6-phosphatase (G6Pase), while the insulin signaling pathway was compromised [106,107,108,109,110]. However, this notion is challenged by a study that showed that the degree of insulin resistance in HCV patients appeared to vary largely due to other risk factors rather than HCV infection [111]. Interestingly, the expression of the bidirectional glucose transporters (GLUT) in HCC can vary among HCC samples. One review [64] cites many studies in which GLUT1 as well as GLUT2 (the low affinity glucose transporter that is preferentially used in the liver for slow uptake or release of glucose) are upregulated in HCC. In contrast, several studies are reported to have shown downregulation of GLUT2 [106,107,108,109]. In the latter case, this condition appears unsuitable for enhanced glucose release from the liver cells, in spite of the fact that gluconeogenesis enzymes are often upregulated. The fact that GLUT2 is downregulated rather argues for the consumption of the carbohydrates produced by gluconeogenesis within the HCC cell itself (e.g. for growth) under conditions of limited release of excess glucose, rather than hormone-regulated release to support the body’s needs for glucose. This would indicate a switch from glucose uptake to the preferential use of amino acids (e.g. glutamine, see above) as carbohydrate backbone sources. We do not know how these contradictory findings relate to the above-mentioned results showing the activation of glycolysis in HCV-infected cancer cells in vitro. We can only speculate that the often observed insulin resistance is a symptom for the tendency of the HCV-infected cells to escape from paramount hormonal control and to reprogram the cell’s metabolism for cancer cell growth, irrespective if glucose or glutamine uptake provides the carbohydrate backbones for growth.

## 4. A Revised View of the Metabolic Conditions in the Warburg Effect

The above considerations about metabolite flux under different conditions in cancer cells—also caused by HCV infection—become even more plausible when taking into account the K_M_ values of the active enzymes. Pyruvate can be converted to Acetyl-Coenzyme A (Acetyl-CoA, Ac-CoA) by the mitochondrial PDH. In the cytosol, it can be converted to alanine by alanine aminotransferase (ALT) and subsequently metabolized to other amino acids. Alternatively, pyruvate can be converted to lactate by LDH. The direction of metabolite flow at this pyruvate junction depends on the activities of the enzymes, on their regulation and on downstream metabolite concentrations.

One of the most important key reactions of carbohydrate catabolism branching from the pyruvate junction is the conversion of pyruvate to Acetyl-CoA by the mitochondrial PDH complex [112]. PDH has a very low K_M_ value (i.e. very high affinity) for pyruvate of about 0.01 mM [113]. Although the turnover rate of the PDH complex is not very high due to the complicated series of reactions in the complex [113], the multimeric PDH complex of about 9 MDa [114] in its fully activated form can be considered to bind pyruvate with very high affinity and efficiently metabolize it to Ac-CoA. The Ac-CoA is then oxidized in the citric acid cycle and the electrons are used in oxidative phosphorylation for ATP production. Although the PDH kinase (PDK) is induced in cancer cells [84,85] (see below), the allosteric feed-back inhibition of PDH activity (indirectly via PDK) by metabolites like NADH and Ac-CoA or its activation by NAD^+^, ADP and pyruvate [70] allows rapid metabolization of pyruvate if required for ATP production.

The second important key reaction branching from the pyruvate junction is the alanine aminotransferase (ALT) reaction. The K_M_ of ALT for the educt pyruvate is about 0.3 mM, and the K_M_ for the product alanine is 28 mM [115,116]. Thus, in cancer cells, the ALT reaction can be regarded to favor the production of alanine from pyruvate, while alanine feedback inhibits the upstream PKM2 activity according to the requirements for the respective amino acids. In contrast, in normal hepatocytes, the inhibition of PK by alanine would save glucose for the brain under conditions when enough amino acids, including alanine, come into the liver from the periphery.

The third key enzyme at the pyruvate junction is lactate dehydrogenase (LDH). The isoform LDHA (i.e. the “M” isoform of LDH, usually expressed in muscle and liver) has a K_M_ value of 0.275 mM for pyruvate, whereas LDHB (i.e. the heart and erythrocyte isoform H) has a K_M_ value of 0.066 mM for pyruvate [117]. In the heart, the low K_M_ of LDHB may serve to rapidly utilize lactate for aerobic energy production, whereas in the erythrocytes, efficient lactate production is required to dispose electrons in the form of lactate in order to ensure ongoing glycolysis in the absence of mitochondria. In metabolically reprogrammed cancer cells, the LDHA isoform with the higher K_M_ for pyruvate was found to be induced even more, while the LDHB isoform was repressed [60]. This may serve to allow flux of pyruvate to lactate only when the pyruvate concentration is high enough to allow the withdrawal of pyruvate for anabolic purposes without shutting down ATP production via PDH, citric acid cycle and oxidative phosphorylation.

A considerable fraction of electrons obtained from the oxidation of carbohydrates in the GAPDH reaction in the cytosol must be imported into the mitochondria for efficient ATP production. For this purpose, essentially, two electron shuttle systems are available. The first electron shuttle system is the malate shuttle [70]. Here, cytosolic malate dehydrogenase (cMDH) transfers the electrons from cytosolic NADH to oxaloacetate and thereby generates malate. The malate is transported into the mitochondrion in exchange for 2-oxoglutarate (α-ketoglutarate). In the mitochondrion, malate is oxidized to oxaloacetate also generating mitochondrial NADH, resulting in the net transfer of the two electrons from cytosolic NADH to mitochondrial NAD^+^. In the cytosol, the 2-oxoglutarate is converted to glutamate in an aminotransferase reaction and the glutamate is transported back to the mitochondrion in exchange to aspartate. The cytosolic aspartate delivers its amino group (e.g. in the urea cycle), regenerating the cytosolic oxaloacetate. The K_M_ of cytosolic MDH is 0.05 mM for oxaloacetate and 0.77 mM for malate, 0.017 mM for NADH and 0.042 mM for NAD^+^ [118], supporting the idea that the malate shuttle is preferentially used for electron transfer from the cytosol to the mitochondrion.

The second electron shuttle system is the glycerophosphate shuttle. The cytosolic glycerol-3-phosphate dehydrogenase (cG3PDH) transfers electrons from cytosolic NADH to glycerone-3-phosphate (dihydroxyacetone-phosphate, DHAP), generating cytosolic glycerol-3-phosphate and reoxidized NAD^+^. Then, the glycerol-3-phosphate is oxidized by the mitochondrial glycerol-3-phosphate dehydrogenase (mG3PDH), which is located on the outside of the inner mitochondrial membrane and transfers the two electrons directly to the ubiquinone pool of the respiratory chain in the inner mitochondrial membrane. The K_M_ of the cytosolic cG3PDH for glycerone-3-phosphate is 0.05 mM, its K_M_ for glycerol-3-phosphate is 0.14 mM, i.e. the K_M_ values are in favor of producing glycerol-3-phosphate. Thus, this glycerophosphate shuttle serves to oxidize cytosolic NADH to NAD^+^ and transfers the electrons to ubiquinone.

Unfortunately, introduction of the electrons from cytosolic NADH directly to ubiquinone but not via the mitochondrial NADH-ubiquinone oxidoreductase (complex I) wastes some ATP. However, under conditions of high glucose consumption, this aspect of energy efficiency may not be so important. Moreover, the inwards transport of electrons into the mitochondrion by the malate shuttle may be hampered by the fact that in cancer cells, the cytosolic aspartate that is required for running the malate shuttle may be withdrawn for anabolic purposes. This argues for the idea that in cancer cells, the glycerophosphate shuttle may be preferentially used to reoxidize the cytosolic NADH.

Reprogramming of the metabolism can be analyzed preferentially by measuring enzyme concentrations. Three interesting studies showed that key enzymes of carbohydrate metabolism are regulated in cancer cells. In a proteomic study, Bentaib and coworkers [60] showed that in cancer cells, several key enzymes of glycolysis are upregulated; this also includes upregulation of PKM2. LDHA is upregulated, while LDHB is downregulated. In contrast, mitochondrial enzymes of the citric acid cycle are downregulated. Moreover, the cytosolic G3PDH was found to be strongly upregulated, while the cytosolic MDH is slightly downregulated [60]. Thus, in cancer cells, electrons from cytosolic NADH that are required for ATP production may be preferentially transported into the mitochondrion by the glycerophosphate shuttle, while the malate shuttle may be hampered by the withdrawal of aspartate for biosyntheses. Interestingly, Bentaib and coworkers also measured the oxygen consumption of cancer cells and found that it is reduced by about 50% in comparison to normal cells [60]. Complementary to the above findings, two other studies [100,101] showed that in HCV-infected hepatocellular carcinoma cells, key enzymes of the pentose phosphate pathway (see Figure 3) are upregulated, like glucose-6-phosphate dehydrogenase (G6PD), phosphogluconate dehydrogenase (PDG), transaldolase (TALDO), and transketolase (TKD), as well as the cytosolic malic enzyme (malate dehydrogenase, decarboxylating, NADP^+^ dependent) that also serves to provide NADPH for biosyntheses. Accordingly, downstream nucleotide synthesis key enzymes are also upregulated [99,100].Liponeogenesis is also upregulated, which, in turn, is essential for feeding HCV assembly via lipid droplets and the VLDL biosynthesis pathway [8,24,25]. To this end, TCA enzymes are only downregulated downstream of citrate synthase [60] to allow flux of citrate to the cytosol as a source for C_2_-units for liponeogenesis [67], while enzymes involved in lipid biosynthesis are upregulated [67,100].

Taken together with the above-mentioned slight downregulation of oxidative phosphorylation [74,80] without shutting it down completely [60], all these changes in cancer cells make sense in order to enhance the production of anabolic metabolites while still running oxidative phosphorylation strongly enough to yield enough energy for growth. Regarding the use versus the disposal of electrons produced in the GAPDH reaction of the highly active glycolysis, a fraction of the cytosolic NADH can transfer its electrons to the mitochondrion for the use in oxidative phosphorylation for efficient ATP production. At the same time, another fraction of NADH can transfer its electrons to that fraction of pyruvate that remains after the withdrawal of C_3_ metabolites from the glycolysis for anabolic purposes. This is—besides the above-mentioned possible bottlenecks in oxygen supply—one important reason why fast-growing cancer cells consume oxygen for oxidative phosphorylation but at the same time release lactate.

In summary, the above findings and considerations suggest a slightly revised view of the metabolic requirements and conditions (so-called Warburg Effect) in cancer cells (Figure 4). The priorities for carbohydrate metabolite flux under this condition are in this order: first, a guarantee for reduced but sufficient ATP production by oxidative phosphorylation; secondly, providing C_6_ and C_3_ metabolites for anabolic purposes; and only thirdly, disposal of “overflow” electrons from excess cytosolic NADH in the form of lactate, which is secreted from the cell. Thus, during cancer cell growth, the production of lactate can be regarded as a collateral event rather than an actual requirement for growth, and it may occur just because oxidative phosphorylation and the withdrawal of upstream metabolites cannot be balanced and fine-tuned in a way that no excess electrons from the GAPDH reaction need to be disposed.

In summary, the above findings and considerations suggest a slightly revised view of the metabolic requirements and conditions (the so-called Warburg Effect) in cancer cells (Figure 4). The priorities for carbohydrate metabolite flux under this condition are in this order: first, a guarantee for reduced but sufficient ATP production by oxidative phosphorylation; secondly, providing C_6_ and C_3_ metabolites for anabolic purposes; and only thirdly, disposal of “overflow” electrons from excess cytosolic NADH in the form of lactate, which is secreted from the cell. Thus, during cancer cell growth, the production of lactate can be regarded as a collateral event rather than an actual requirement for growth, and it may occur just because oxidative phosphorylation and the withdrawal of upstream metabolites cannot be balanced and fine-tuned in a way that no excess electrons from the GAPDH reaction need to be disposed.

## 5. HCV Induces Fibrosis and Cancer Growth

In addition to reprogramming the metabolism of the infected cell, HCV also induces many gene expression changes in the cells that drive the cells towards the establishment of cancer, which have been detailed in excellent reviews [58,73,119,120,121]. Here, we focus on some transcriptome changes that emerged from the experiments of our previous study [74] but were not detailed before. KEGG (Kyoto Encyclopedia of Genes and Genomes) pathway analyses [122] using GAGE software (v2.34.0) [123] of cellular transcriptome changes in hepatocellular carcinoma cells that replicate HCV for 6 days [74] revealed osteoclast differentiation, MAPK signaling, TGF-β signaling, and retinoate metabolism as the four most important upregulated pathways (see Figure 5).

It is unclear whether the activation of osteoclast differentiation by HCV and the resulting increase in serum Ca^2+^ concentration is somehow related to the role of increased Ca^2+^ concentrations in the cytosol and in mitochondria during the stress response activated by HCV infection [124,125]. Similarly, it is not clear whether the HCV infection itself or the events involved in development of fibrosis induced by the HCV infection are related to the increase in the risk of osteoporosis. However, abnormal bone metabolism that results in low bone mineral density (BMD) and osteoporosis is known to be associated with liver disease and HCV infection correlates with events related to osteoclast differentiation. As a general finding, the risk of osteoporosis was approximately 65% higher among HCV-infected patients compared with those without HCV infection, and HCV infection was speculated to be an independent risk factor for post-menopausal BMD loss and fractures [126,127,128]. Cirrhosis patients have been shown to have increased expression of the receptor-activator ratio of nuclear factor kappa ligand (RANKL) as well as of osteoprotegerin (OPG), which results in increased bone resorption. In addition, liver inflammation caused by HCV infection modulates the bone-remodeling pathway through pro-inflammatory cytokines such as interleukin (IL)-1, IL-6, and IL-17 and tumor necrotic factor (TNF)-α, which may promote the development of osteoclasts.

We found that the MAPK signaling pathway as well as the transcription factor c-Jun are up-regulated in HCV replicating HCC cells [74]. Moreover, Deng and coworkers found that c-Jun expression and the MAPK pathway are activated in HCV infected and HCC patients, and Jun activation resulted in cell cycle progression [129]. c-Jun and STAT3 are widely accepted critical regulators of liver cancer development and progression. We recently demonstrated that activation of c-Jun and STAT3 as well as DNA repair were also induced by an extract from schistosome eggs [130]. Clinical studies demonstrated that the coinfection of HBV and HCV in combination with S. mansoni aggravate the clinical course of hepatitis but also of hepatocellular carcinogenesis [131]. Thus, the permanent activation of c-Jun and STAT3 as critical regulators in liver cancerogenesis could directly influence hepatocarcinogenesis and chronic HCV infection might reinforce these mechanisms. Moreover, cell cycle progression is promoted by the NS5B protein of HCV which binds to the retinoblastoma protein (Rb) and promotes its degradation [73,132].

Accordingly, the HCV envelope protein E2 was shown to specifically activate the MAPK/ERK pathway via its receptors and to greatly promote cell proliferation [133]. Intracellularly, the HCV core protein enhances cell proliferation by inhibiting the synthesis of tumor suppressor p53, the downstream p21 CDK inhibitor and E2F-1, and the HCV core protein induces the phosphorylation of the tumor suppressor Retinoblastoma protein (pRb) [134]. Thus, HCV activates growth factor-related signaling pathways which promote cancer cell growth, while pathways involved in the possible activation of apoptosis are rather inhibited.

The signaling pathway of transforming growth factor beta (TGF-β) was also activated during HCV replication [74]. TGF-β is a multifunctional profibrotic cytokine that plays a key role in the pathogenesis of liver inflammation, fibrosis, cirrhosis and HCC [135]. Members of the TGF-β family control numerous cellular functions including proliferation, apoptosis, differentiation, epithelial-mesenchymal transition (EMT) and migration. In early stages of cancer, TGF-β exhibits tumor suppressive effects by inhibiting cell cycle progression and promoting apoptosis. However, in late stages of cancer, TGF-β exerts tumor promoting effects, increasing tumor invasiveness and metastasis. Elevated TGF-β activity has been associated with poor clinical outcome [136]. HCV-infected hepatocytes were reported to release TGF-β1 and other profibrogenic factors that differentially modulate the expression of several key genes involved in liver fibrosis in hepatic stellate cells (HSCs) [137]. Quiescent HSCs are known as vitamin-A-storing cells in the liver. Moreover, during injury, they are activated and become proliferative, fibrogenic and contractile myofibroblasts and now are well established as a central driver of fibrosis [71]. Thus, elevated TGF-β expression in HCV-replicating cells contributes to fibrosis.

All-trans-retinoic acid (“retinoate” in the following) is involved in the regulation and promotion of differentiation [138]; cancer cells in which CYP26A1 expression was suppressed had reduced tumorigenicity [139]. Thus, a decrease in retinoate levels may be an indicator for a tendency of cells to approach or maintain a less differentiated state (like cancer cells). As a consequence, the loss of hepatic retinoic acid function leads to the development of steatohepatitis and liver tumors [140], whereas the application of retinoate reduced the amount of histologically detectable fibrosis and oxidative stress [141].

In KEGG analyses of our previous transcriptome sequencing data from HCV replicating hepatoma cells, we found that genes involved in the pathway for inactivation of retinoate are induced [74]. This particularly includes CYP26A1, which is essential for the regulation of embryonic development [139]. Because retinoate can easily distribute in the body, the regulation of its synthesis as well of its degradation are key for its activity. Misregulation of retinoate concentrations can cause severe distortions of embryonic development and stem cell differentiation; therefore, too low as well as too high concentrations of retinoate may cause similar defects [138]. As a general rule, retinoate can be regarded to be required for proper differentiation. As a consequence, enhanced inactivation of retinoate, e.g. by CYP26A1, may result in less differentiation of cells and tissues [139]. This notion is consistent with the finding that HCV manipulates the infected cells to be in a less differentiated state and may contribute to the development of hepatocellular carcinoma [74].

## 6. Conclusions

Although HCV comes with a genome of less than 10 kilobases in length, HCV infection of hepatocytes induces a wide variety of changes in cellular gene expression and regulation in the infected cells. These changes include early programming of the infected cells to develop cancer, and metabolic switches are induced in the cancer cells that results in enhanced metabolite flux through glycolysis to allow for increased production of anabolic metabolites while still producing enough ATP by oxidative phosphorylation. In contrast, even though results can be variable, in HCC, patients often an upregulation of key enzymes of gluconeogenesis can be found, with concurrent downregulation of the GLUT2 transporter that would be required for glucose release from the cancer cell. This suggests that whatever condition occurs, it must be suspected not to fulfill the body’s needs for glucose release from the liver, but the glucose must be suspected to be produced for cancer cell growth.

In conditions of enhanced glycolysis, the increased lactate release from the cancer cells may be regarded largely as a collateral damage rather than a strict metabolic requirement for cancer cell growth. This lactate release may occur mainly because the cells are not able to fine-tune and balance anabolic metabolite production and pyruvate consumption by the TCA cycle and oxidative phosphorylation so accurately that electron disposal in the form of lactate in order to provide reoxidized NAD^+^ for running the GAPDH reaction in glycolysis could be completely avoided.

## Figures and Tables

**Figure 1 cells-08-01410-f001:**
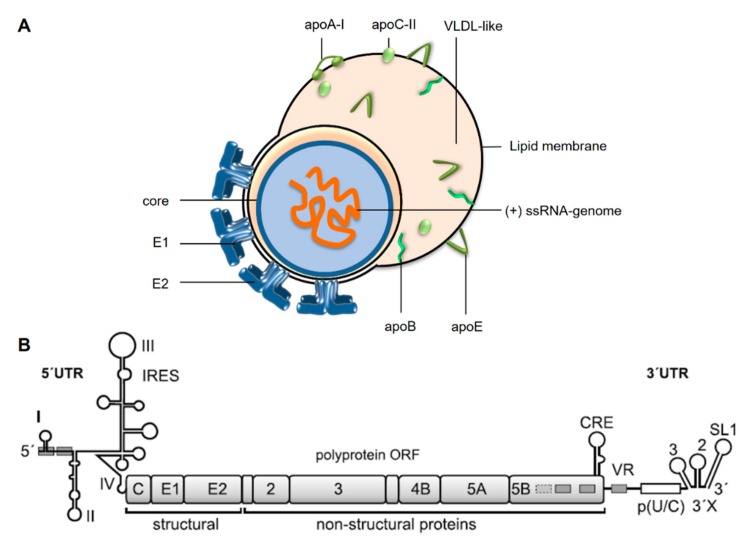
Hepatitis C Virus (HCV). (**A**) The HCV particle comes as a fusion lipo-viro particle associated with components of very low-density lipoprotein (VLDL) particles [8]. The single-stranded HCV RNA genome of positive polarity is covered by the core protein. In the host cell-derived lipid membrane, the HCV envelope proteins E1 and E2 are localized. The HCV lipo-viro particle dynamically acquires various amounts of lipids to additionally form a VLDL-like portion of the fusion particle, which is associated with the apolipoproteins (apo) B, E, A-I and C-II. (**B**) The HCV genome of about 9600 nucleotides encodes a single polyprotein open reading frame (ORF) which is co- and post-translationally processed into the mature gene products, including the structural core and envelope proteins and the non-structural (NS) proteins of the replication complex. The NS5B protein constitutes the viral replicase, an RNA-dependent RNA polymerase. The 5′- and 3′-untranslated regions (UTRs) harbor the sequences and RNA secondary stem-loop (SL) structures that are involved in the regulation of viral genome translation and replication; in the 5′UTR, these stem loops are numbered with roman numerals. The actual AUG start codon is part of SL IV. The SLs II to IV constitute the Internal Ribosome Entry Site (IRES). In the 3′UTR, the highly conserved so-called 3′X region contains SLs 1 to 3 (numbered from 3′ to 5′), preceded by a poly(U/C) tract and a variable region (VR). Regulation of viral genome replication requires several additional signals also in the coding regions, prominently represented here by the cis-replication element (CRE). Binding sites in the 5′UTR, the NS5B coding region and the VR for the liver-specific microRNA-122 are represented by grey boxes. For more details, please refer to [12] and [13].

**Figure 3 cells-08-01410-f003:**
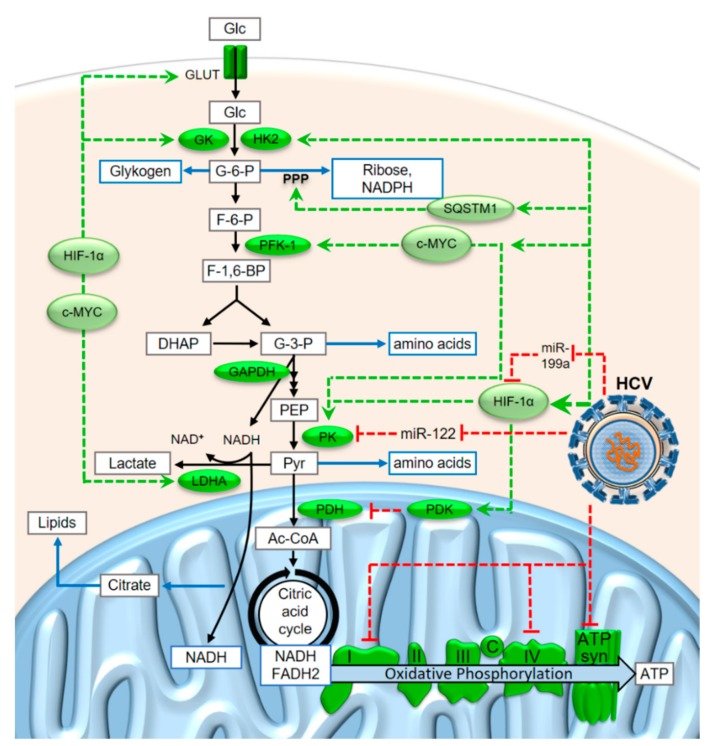
Metabolite flux through glycolysis and oxidative phosphorylation and some of the modifications induced in cancer cells and during HCV replication. Abbreviations: Glc, glucose; G-6-P, glucose-6-phosphate; F-6-P, fructose-6-phosphate; F-1,6-BP, fructose-1,6-bisphosphate; DHAP, dihydroxyacetone-phosphate (= glyceron-3-phosphate); G-3-P, glyceral(dehyde)-3-phosphate; PEP, phosphoenolpyruvate; Pyr, pyruvate; Ac-CoA, Acetyl-Coenzyme A; GLUT, glucose transporter; GK, glucokinase; HK2, hexokinase 2; PFK-1, phosphofructokinase-1; PK, pyruvate kinase, LDHA, lactate dehydrogenase A; SQSTM1, Sequestosome 1; PDH, pyruvate dehydrogenase; PDK, PDH kinase; c, cytochrome c; ATP syn, ATP synthase. More explanations are given in the main text.

**Figure 4 cells-08-01410-f004:**
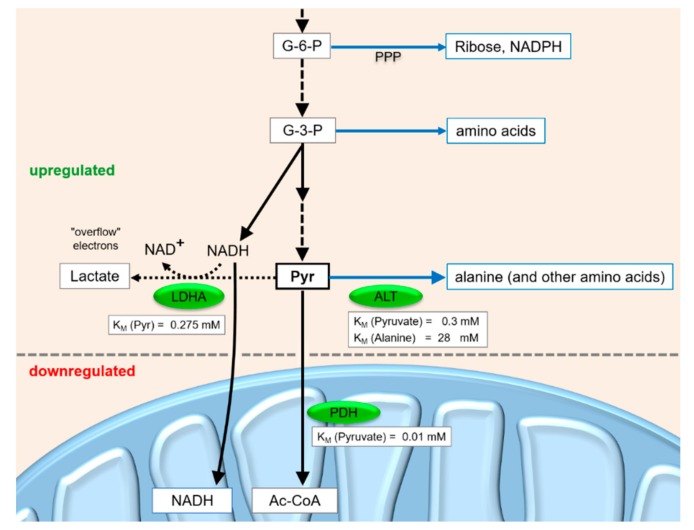
Metabolite flux at the pyruvate junction in fast-growing cancer cells.

**Figure 5 cells-08-01410-f005:**
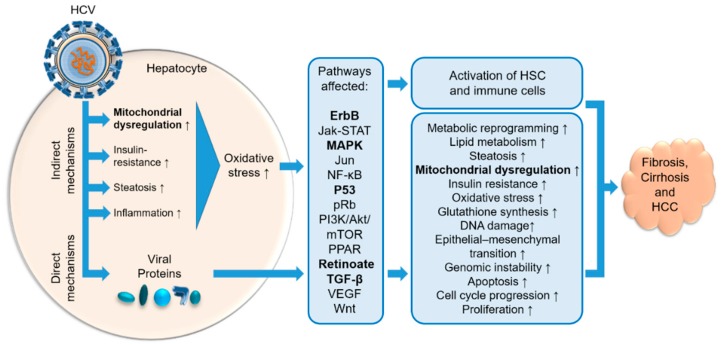
The interplay between HCV infection and changes in cellular gene expression and metabolic programming, leading to liver fibrosis, cirrhosis and cancer. Pathways which were found to be affected by HCV replication in our previous study [74] are marked in bold. For details, please see the main text.

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
