# Peer review of "Hepatitis C Virus Downregulates Core Subunits of Oxidative Phosphorylation, Reminiscent of the Warburg Effect in Cancer Cells"

_cells, 2019, doi:10.3390/cells8111410_

Round 1
Reviewer 1 Report
cells-630994
Title: Hepatitis C Virus replication induces downregulation of core subunits of oxidative phosphorylation to support the Warburg Effect for cancer cell growth
This paper reviewed chronic viral infection is closely associated with liver fibrosis, cirrhosis and hepatocellular carcinoma. In addition, HCV downregulates the expression of mitochondrial oxidative phosphorylation complex core subunits quite early after infection. HCV deregulates signaling pathways like those of TNF-β and MAPK by direct and indirect mechanisms, which can lead to fibrosis and HCC. This idea is interesting and well supported with reasonable data in the manuscript. However, I have the following comments to improve this manuscript. I find your review paper interesting and with merit for publication, but before I can reach a final decision, I need some clarification:
Minor
As you know, viruses replicate in different cell types and under different physiological conditions of their host cells. In some cells a given virus performs an efficient lytic infection producing a big load of viral progeny, whereas in other cells it may carry out a long-lasting eventually lifelong persistence (asymptomatic latent infection) or a persistence with recurrent symptomatic infection. Therefore, authors may be described the metabolic pathways under these conditions. The metabolic reprogramming often leads in the infected host cells to enhanced glucose uptake, aerobic glycolysis, production and secretion of lactate together with reduced activity of the TCA and OXPHOS. In some cases, Glutamine may serve as additional or predominant carbon substrate which replenishes the TCA through glutaminolysis. Glutamine is essential in many aspects of cell metabolism. After entering the cell through specific transporters, glutamine can be partially oxidized through glutaminolysis. What is the glutaminolysis metabolic pathway after cells infection with HCV. The induced catabolism allows activation of anabolic pathways necessary for the production of the viral nucleic acids, capsids, and eventually membrane envelopes. Viruses pursue different strategies to meet these metabolic requirements, but most viruses interact at some point during their replication cycle with the PI3K/Akt/mTOR pathway through binding of viral factors to catalytic subunit of PI3K in order to inhibit host cell death and/or to modulate cellular metabolism. Authors might be clearly added the PI3K/Akt/mTOR pathway after infection with HCV Glutamine is essential in many aspects of cell metabolism. After entering the cell through specific transporters, glutamine can be partially oxidized through glutaminolysis. A major limitation in current understanding of viral-induced metabolic reprogramming stems from the fact that most of the work characterizing viral alterations to host cell metabolism so far has been carried out in vitro. However, metabolism in vivo is known to be quite different from that found in vitro in cell culture conditions. Therefore, authors described in vivo models to assess metabolic changes induced by virus infection are needed to have a more accurate understanding of viral metabolism and facilitate therapeutic antiviral strategies.
Author Response
Reviewer 1
This paper reviewed chronic viral infection is closely associated with liver fibrosis, cirrhosis and hepatocellular carcinoma. In addition, HCV downregulates the expression of mitochondrial oxidative phosphorylation complex core subunits quite early after infection. HCV deregulates signaling pathways like those of TNF-β and MAPK by direct and indirect mechanisms, which can lead to fibrosis and HCC. This idea is interesting and well supported with reasonable data in the manuscript. However, I have the following comments to improve this manuscript. I find your review paper interesting and with merit for publication, but before I can reach a final decision, I need some clarification:
Minor
As you know, viruses replicate in different cell types and under different physiological conditions of their host cells. In some cells a given virus performs an efficient lytic infection producing a big load of viral progeny, whereas in other cells it may carry out a long-lasting eventually lifelong persistence (asymptomatic latent infection) or a persistence with recurrent symptomatic infection. Therefore, authors may be described the metabolic pathways under these conditions. The metabolic reprogramming often leads in the infected host cells to enhanced glucose uptake, aerobic glycolysis, production and secretion of lactate together with reduced activity of the TCA and OXPHOS.
Reviewer comment:
In some cases, Glutamine may serve as additional or predominant carbon substrate which replenishes the TCA through glutaminolysis. Glutamine is essential in many aspects of cell metabolism. After entering the cell through specific transporters, glutamine can be partially oxidized through glutaminolysis. What is the glutaminolysis metabolic pathway after cells infection with HCV.
…
Glutamine is essential in many aspects of cell metabolism. After entering the cell through specific transporters, glutamine can be partially oxidized through glutaminolysis.
Reply:
Thank you very much for this very helpful comment! We have now included a section on glutamine and glutathione metabolism in HCV infected cells in chapter 3, explaining that glutamine metabolizing and glutathione producing enzymes are upregulated in HCV infected cells. The use of glutamine also fits well with the switch to gluconeogenesis conditions (instead of glycolysis) with at the same time downregulated GLUT2 (see below at in vivo conditions).
Reviewer comment:
The induced catabolism allows activation of anabolic pathways necessary for the production of the viral nucleic acids, capsids, and eventually membrane envelopes. Viruses pursue different strategies to meet these metabolic requirements, but most viruses interact at some point during their replication cycle with the PI3K/Akt/mTOR pathway through binding of viral factors to catalytic subunit of PI3K in order to inhibit host cell death and/or to modulate cellular metabolism. Authors might be clearly added the PI3K/Akt/mTOR pathway after infection with HCV.
Reply:
Thank you very much for this suggestion! We have now added Akt/mTOR to the PI3K pathway noted in Fig. 5, this figure was extended also due to some other Reviewer suggestions. Moreover, the Akt/mTOR pathway was mentioned in the text in section 3.
Reviewer comment:
A major limitation in current understanding of viral-induced metabolic reprogramming stems from the fact that most of the work characterizing viral alterations to host cell metabolism so far has been carried out in vitro. However, metabolism in vivo is known to be quite different from that found in vitro in cell culture conditions. Therefore, authors described in vivo models to assess metabolic changes induced by virus infection are needed to have a more accurate understanding of viral metabolism and facilitate therapeutic antiviral strategies.
Reply:
Thank you very much for this valuable comment! We have now discussed the contradictory results emerging from in vivo models but in particular from data obtained from HCC patients in a new paragraph at the end of section 3, including many new citations. These findings also fit well with enhanced glutaminolysis (see above). Moreover, we have slightly modified the conclusion section accordingly. However, we considered it too speculative to discuss a possible direct link from the above findings about metabolic changes to therapeutic antiviral strategies, therefore, we did not discuss this point, since direct acting antivirals are quite efficient.
Reviewer 2 Report
Present study outlines a comprehensive view of published information about the underlying mechanisms of the role of hepatitis C virus (HCV) infection to development of liver diseases. Although, not much has been mentioned about hepatocellular carcinoma (HCC) by HCV. I would like to recommend authors to draw a schematic or table for the role of HCV in HCC and the cell cycle proteins involved in it. Otherwise, the paper looks great i am recommending for its acceptance which would acquire lot of citations.
Author Response
Reviewer 2
Present study outlines a comprehensive view of published information about the underlying mechanisms of the role of hepatitis C virus (HCV) infection to development of liver diseases. Although, not much has been mentioned about hepatocellular carcinoma (HCC) by HCV. I would like to recommend authors to draw a schematic or table for the role of HCV in HCC and the cell cycle proteins involved in it. Otherwise, the paper looks great i am recommending for its acceptance which would acquire lot of citations.
Reply:
Thank you very much for this suggestion. In Figure 5, the contribution of HCV to HCC development ist illustrated. We have now included in Figure 5 also Jun activation, Rb inactivation as well as the activation of Cell cycle progression by HCV. An explanation that cell cycle progression was activated was added in the text in the context of the Deng 2019 study and the MAPK pathway.
Reviewer 3 Report
Gerresheim et al. wrote a review on metabolic shift after HCV infection. Based on many reports and the authors’ recent discovery of reduced expression of genes related with mitochondrial oxidative phosphorylation complex core subunits, the authors drew an analogy between HCV infection and cancer generation. The authors’ comprehensive analyses and explanations provide excellent insight into the global changes in cellular metabolism in HCV-infected cells.
As the authors suggested, the enhanced glycolysis flux by HCV infection is likely to provide anabolic metabolites for the syntheses of HCV RNAs and proteins supporting HCV proliferation. However, the relationship between the metabolic shift by HCV infection and cancer development and/or cancer cell growth is not clear as implied by the title of the paper.
Importantly, the authors may describe in detail about the potential contribution of the metabolic shift to lipogenesis (Please refer to Costello and Franklin (2005)). The Warburg effect augments lipogenesis, and HCV protein(s) are known to induce lipogenesis. Moreover, it is now clear that enhanced lipogenesis plays a key role in HCV RNA replication, encapsidation, and virus secretion.
Minor point
HCV IRES does not have weakest IRES activity. CrPV IRES is weaker.
Author Response
Reviewer 3
Gerresheim et al. wrote a review on metabolic shift after HCV infection. Based on many reports and the authors’ recent discovery of reduced expression of genes related with mitochondrial oxidative phosphorylation complex core subunits, the authors drew an analogy between HCV infection and cancer generation. The authors’ comprehensive analyses and explanations provide excellent insight into the global changes in cellular metabolism in HCV-infected cells.
As the authors suggested, the enhanced glycolysis flux by HCV infection is likely to provide anabolic metabolites for the syntheses of HCV RNAs and proteins supporting HCV proliferation.
Reviewer comment:
However, the relationship between the metabolic shift by HCV infection and cancer development and/or cancer cell growth is not clear as implied by the title of the paper.
Reply:
Thank you very much for this comment; we have now corrected the title in a way that a proven strict correlation is not suggested.
Reviewer comment:
Importantly, the authors may describe in detail about the potential contribution of the metabolic shift to lipogenesis (Please refer to Costello and Franklin (2005)). The Warburg effect augments lipogenesis, and HCV protein(s) are known to induce lipogenesis. Moreover, it is now clear that enhanced lipogenesis plays a key role in HCV RNA replication, encapsidation, and virus secretion.
Reply:
Thank you very much for this comment! We have now included the export of citrate from the mitochondrion to the cytosol in the drawing in Fig. 3. Interestingly, the notion of Costello & Franklin fits perfectly together with the finding of the Bentaib proteomics study that citrate synthase is not downregulated but TCA cycle enzymes only downstream of citrate synthase are downregulated in cancer cells, in order to allow flux of citrate to the cytosol for lipid synthesis - which in turn is required for HCV assembly. In addition, Sugiyama et al. showed that lipid biosynthesis genes are upregulated. This perfect match is now explained in the text in section 4.
Minor point
HCV IRES does not have weakest IRES activity. CrPV IRES is weaker.
Reply:
Thank you very much, we have corrected this to "is quite weak compared to cap-dependent translation".